# stoPET v1.0: A stochastic potential evapotranspiration generator for simulation of climate change impacts

Dagmawi Teklu Asfaw[1], Michael Bliss Singer[2,3,4], Rafael Rosolem[5,6], David MacLeod[1],
Mark Cuthbert[2,7], Edisson Quichimbo Miguitama[2], Manuel F. Rios Gaona[2], Katerina Michaelides[1,4,6]

[1] School of Geographical Sciences, University of Bristol, Bristol, UK
[2] School of Earth and Environmental Sciences, Cardiff University, Cardiff, UK
[3] Water Research Institute, Cardiff University, Cardiff, UK
[4] Earth Research Institute, University of California Santa Barbara, Santa Barbara, USA
[5] Department of Civil Engineering, University of Bristol, UK
[6] Cabot Institute for the Environment, University of Bristol, Bristol, UK
[7] School of Civil and Environmental Engineering, The University of New South Wales (UNSW), Sydney, Australia

*Correspondence to*: Dagmawi Teklu Asfaw (d.t.asfaw@bristol.ac.uk)

**Abstract.** Potential evapotranspiration (PET) represents the evaporative demand in the atmosphere for the removal of water from the land and is an essential variable for understanding and modelling land-atmosphere interactions. Weather generators are often used to generate stochastic rainfall time series; however, no such model exists for stochastically generating plausible PET time series. Here we develop a stochastic PET generator, stoPET, by leveraging a recently published global dataset of hourly PET at 0.1° resolution (hPET). stoPET is designed to simulate realistic time series of PET that capture the diurnal and seasonal variability of hPET and to support the simulation of various scenarios of climate change. The parsimonious model is based on a sine function fitted to the monthly average diurnal cycle of hPET, producing parameters that are then used to generate any number of synthetic series of randomised hourly PET for a specific climate scenario at any point of the global land surface between 55° N and 55° S. In addition to supporting stochastic analysis of historical PET, stoPET also incorporates three methods to account for potential future changes in atmospheric evaporative demand to rising global temperature. These include 1) user-defined percentage increase of annual PET; 2) a step change in PET based on a unit increase in temperature, and 3) extrapolation of the historical trend in hPET into the future. We evaluated stoPET at a regional scale and at twelve locations spanning arid and humid climatic regions around the globe. stoPET generates PET distributions that are statistically similar to hPET and an independent PET dataset from CRU, capturing their diurnal/seasonal dynamics, indicating that stoPET produces physically plausible diurnal and seasonal PET variability. We provide examples of how stoPET can generate large ensembles of PET for future climate scenario analysis in sectors like agriculture and water resources, with minimal computational demand.

## 1    Introduction

Potential evapotranspiration (PET) is the representation of the atmospheric demand for evaporation from a well-watered, vegetated land surface (Allen et al., 1998). It is paramount in determining the water balance within hydrological models and is routinely used in water management for agriculture to determine crop water demand and irrigation scheduling. PET is also a crucial input in climate change impact studies which, for example, aim to provide actionable information on water scarcity (Raziei and Pereira, 2013; Liu et al., 2019; Tasumi, 2019; Zhou et al., 2020; Quichimbo et al., 2021). However, the estimation of PET is limited by the availability and quality of meteorological data at the spatial and temporal resolution appropriate to the purpose of a given study and by uncertainty in future climate. Differences between PET calculation methods influence the output of hydrological models, so the ability to simulate multiple realisations of PET under different scenarios of climate change via a single estimation method is vital to quantify uncertainties in the water balance due to changes in evaporative demand from the atmosphere (Valipour et al., 2017a; Dallaire et al., 2021). In studies that compare different methods of PET

estimation (Tukimat et al., 2012; Li et al., 2016; Valipour et al., 2017b), the Penman-Monteith (PM) equation is used as a reference against which other methods are compared. Though the PM equation is the most common and accepted method of choice for PET estimation, it is highly data intensive, requiring many input variables (Allen et al., 1998; Grismer et al., 2002; Mohawesh 2011; Ravazzaniv et al., 2012; Lee and Cho, 2012; Tukimat et al., 2012). This limits its utility and relevance, particularly for the many data-sparse regions across the globe (Yadeta et al., 2020). The lack of adequate local meteorological data necessitates reliance on empirical methods of PET estimation, which require intensive calibration (Kingston et al., 2009), and can in turn limit the accuracy of resulting PET products.

While some global climate models do not include PET explicitly (e.g., COSMO-CLM (Will et al., 2017)), most global climate models (e.g., ERA5-Land) do provide some of the outputs of climatic variables used to estimate PET. However, they do not directly output PET itself, which would support more detailed impact-based modelling of climate change. Climate models focus on predicting the effects of greenhouse gas emissions on global water and energy transfer, and thus they output climate variables (e.g., temperature, radiation, surface pressure, wind speed, and rainfall). Without explicit data on PET, high computational resources are required to estimate the PET for large areas from climate model output variables, and the spatial and temporal scales of these outputs are typically too coarse for detailed impact analyses. These scaling considerations may make climate model output unsuitable for computing PET. This is especially true for application to certain water balance applications, where diurnal changes in PET are important for a specific location or where there are large spatial differences in PET. Downscaling techniques are commonly used to generate the parameters needed to estimate PET from global climate models by the PM method (or other methods) at the appropriate resolution, but this increases the computational resource requirement (Tukimat et al., 2012) and adds additional uncertainty to PET calculations.

Another challenge for PET estimation is how to characterise evaporative demand under climate change scenarios, which is an important need for assessing possible future climate change impacts (Xu et al., 2014). Temperature is one of the major climate variables influencing PET (Allen et al., 1998). Therefore, with increasing temperature under climate change for most of the globe, there is a need to simulate historical and future PET in a consistent and spatially explicit way. Simulating changes in evaporative demand associated with changes in temperature would be particularly useful for assessing the potential impacts of meeting/not meeting the 1.5° C target of the Paris Climate Treaty (Kriegler et al., 2018) or for addressing any future global temperature target. Additionally, it would be powerful to be able to simulate step changes and trends in PET according to user-defined specifications, giving the user a flexible tool for generating a range of PET time series for various applications.

Given the inherent uncertainty in climatic drivers on the terrestrial water balance and the need to incorporate current and future PET trends in hydrological and other climate change impact models, stochastic PET simulation provides a flexible and useful tool to fill this research gap. While several stochastic weather generators exist and are used to generate physically consistent time series of rainfall (Fatichi et al., 2011; Peleg et al., 2017; Singer et al., 2018; De Luca et al., 2020), no similar model exists for generating stochastic PET time series. Although PET calculations are sometimes included within hydrological models, these require user specification of input climate variables used in the calculation and a specification of the calculation method. In these cases, PET is internally calculated to close the water balance, but it is not typically provided as an output variable. Ultimately, there is no existing method for obtaining internally consistent simulations of PET at high spatial and temporal resolution for the entire global land surface. This paper addresses this gap and introduces a new stochastic PET generator, stoPET, for simulating hourly time series of PET at 0.1° spatial resolution for the global land surface. stoPET enables the user to characterise the uncertainty in PET for historical and future climate scenarios. It supports the generation of unlimited unique realisations of PET in a computationally efficient way. To support analyses of climate change, stoPET incorporates different methods to account for potential changes in atmospheric evaporative demand in response to rising global temperature, supporting flexibility in simulating various climate scenarios. The importance of including options to simulate multiple future

PET time series emanates from the unpredictability of future climate and the need to assess the impacts of climatic changes on the water balance.

Below we provide a comprehensive description of the stoPET model and its potential application for predicting the evolution of water resources in drylands, estimating future crop water demand, assessing flash flood potential, or providing actionable information on expected climatic impacts on the water balance. Section 2 describes the concept and design of the model with a brief note about its implementation. Section 3 describes the model verification at regional and point scale. Section 4 describes the methods used to incorporate PET changes due to temperature changes in the stoPET model. The paper concludes with a discussion of the potential application of stoPET (Sect. 5). A user manual for stoPET is included as a supplement, and all the model scripts and input parameters are freely available on figshare ([10.6084/m9.figshare.19665531](10.6084/m9.figshare.19665531)).

## 2 Model concept and design

### 2.1 Concept

The stoPET model generates hourly PET values based on sine function parameters estimated from hPET (Singer et al., 2021), an hourly PET dataset that was recently created from ERA5-Land climatic variables (Muñoz Sabater, 2019) using the Penman-Monteith method (Allen et al., 1998). The resulting PET generated from stoPET retains the diurnal and seasonal variations in PET contained within the hPET dataset, but notably, stoPET injects randomness (stochasticity) in the simulated series via a noise factor. In other words, stoPET does not recreate hPET, but rather uses hPET to generate new randomised sequences of PET based on the diurnal and seasonal variability in the hPET dataset. The development of stoPET begins by using the entire hPET dataset as an input, from which we create a generalised functional form for diurnal PET and create a noise factor to inject stochasticity according to the following steps, each of which is outlined in more detailed in subsequent sections below:

1) Estimate the average diurnal cycle of PET for each month using a sine function
2) Fit a skewed normal distribution to the difference between all hourly values for the diurnal curve and the average diurnal curve, for of each month to generate a randomized noise ratio.
3) Generate stochastic PET timeseries for a particular month by multiplying that month's average diurnal cycle with a sequence of draws from the corresponding skewed normal distribution

### 2.2 Model implementation

The overall stochastic PET generation model, stoPET, can be expressed as follows:

Stochastic PET = (Average diurnal cycle of PET using a sine function * a random Noise ratio) +
user defined annual PET variability

Each of the three components is described in detail in the subsequent sections.

### 2.2.1 Sine function parameter estimation

The stoPET model is based on fitting a sine function to the average diurnal cycle calculated from hPET for each month and for each grid cell. The sine function, defined in Eq. (1), provides the four parameters required to represent the characteristic of hourly PET for each month at each grid cell:

$$Y = A \, sin \, (B * t \, + \, C) \, + D \qquad\qquad \text{Eq. (1)}$$

Where $A$ represents the diurnal amplitude (mm h$^{-1}$), $B$ is the frequency (h$^{-1}$), $C$ is the phase shift (-), $D$ is the vertical shift (mm h$^{-1}$). $t$ is time (h), and $Y$ is the new PET value (mm h$^{-1}$) generated from the sine function.

The monthly sine fit is based on the average of values of hPET for all diurnal curves for all days of that month over the period of record (for this application, 1981-2020). The sine fit is only done based on values for daylight hours (sunrise to sunset), as we assume nighttime PET values are zero. In reality, PET is not always zero at night, but it typically ranges from small positive to small negative values (representing condensation) within hPET. For example, nighttime PET is relatively higher in arid regions (median PET value is between 0.001 and 0.076 mm h$^{-1}$) compared to humid regions (median PET value is between -0.014 and 0.002 mm h$^{-1}$) (Fig. S1 and Fig. S2). Nevertheless, the impact of nighttime PET in core applications such as crop and hydrological modelling is expected to be minimal, hence we set nighttime PET values to zero in stoPET.

An example of the sine function representing hPET data for a single grid location (Wajir in Kenya - 1.73° N, 40.09° E) for the month of January is shown in Fig. 1. The grey shaded area represents the range of the hourly PET obtained from all days of January within the 40 years record of hPET data, while the black dotted line shows the average of those hPET values. This average diurnal cycle is used to fit the sine function (red solid line based on $Y$ in Eq. (1)) for each month of the year. The four parameters from Eq. (1) are estimated at each 0.1° grid location for each month and then saved as input for simulating synthetic sequences of PET. Figure 2 shows, for illustration, the spatial variability of parameters across the globe for January. For each month of the year all four parameters, plus the sunrise and sunset hours (which are required to identify daytime and nighttime periods) for any pixel across the global land surface (Figure 2), are provided as an input file to be run with the model script.

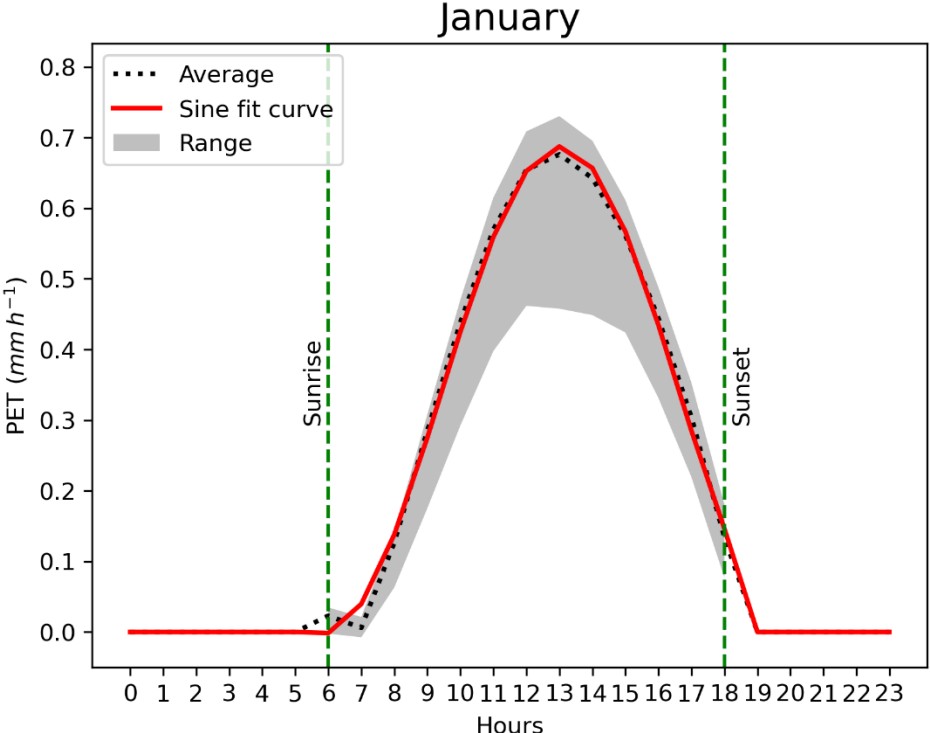

**Figure 1: An example of a sine function curve fitted over the average hourly PET values for January at a location in Wajir (Kenya). The black dotted line is the average from hPET, and the red solid line represents the fitted sine function. The grey shaded area is the range across all January days in the 40 years of record for hPET. Average sunrise and sunset times are shown in green vertical dashed lines.**

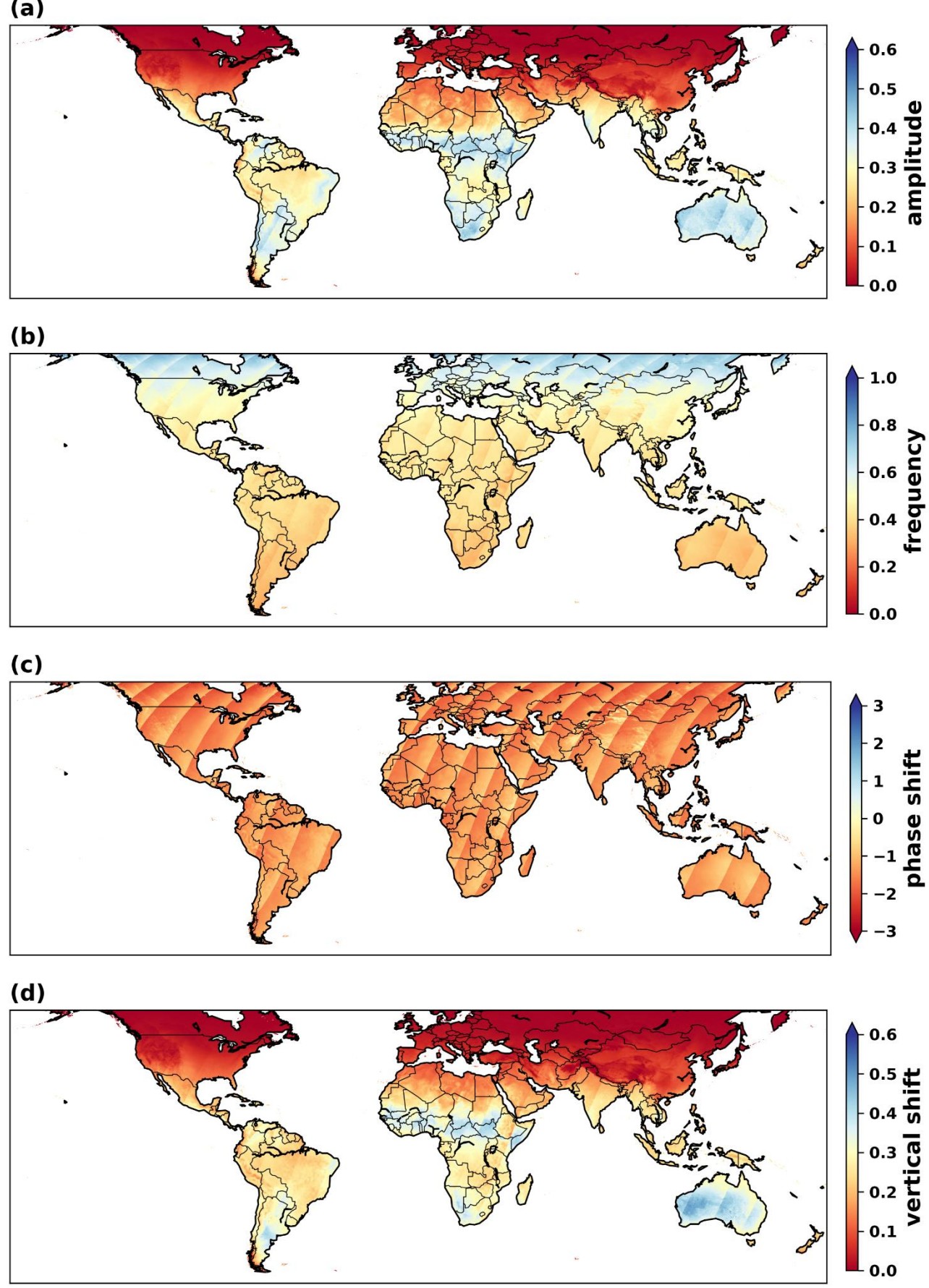

**Figure 2: The sine function parameters estimated for January over the spatial domain of the stoPET model (global land surface latitudes between 55° N and 55° S). The parameters are described in Eq. (1) where (a) the amplitude (mm h⁻¹), (b) the frequency (h⁻¹), (c) the phase shift (-) and (d) the vertical shift (mm h⁻¹).**

#### 2.2.2 Random noise estimation

PET shows variability within each month (Fig. 1), which is represented stochastically in stoPET using a "noise ratio" parameter ($N$) (Eq. (2)):

$$N(h, d, m) = \frac{PET(h,d,m)}{\overline{PET}(h,m)}$$

Eq. (2)

Where $PET\ (h, d, m)$ is the PET for every hour (h) and day (d) of each month (m) and $\overline{PET}\ (h, m)$ is the average PET of each hour over all days of the month. A skewed normal distribution is then fitted to noise ratios of each month calculated using Eq. (2). The fitted skewed normal distribution parameters (skewness, location, and scale), defined at each grid cell and month, are used as input to stoPET to generate stochastic variability around the sine function by sampling from this skewed distribution. Figure 3 shows the values of the three noise ratio parameters over the entire spatial domain of stoPET, estimated for the month of January.

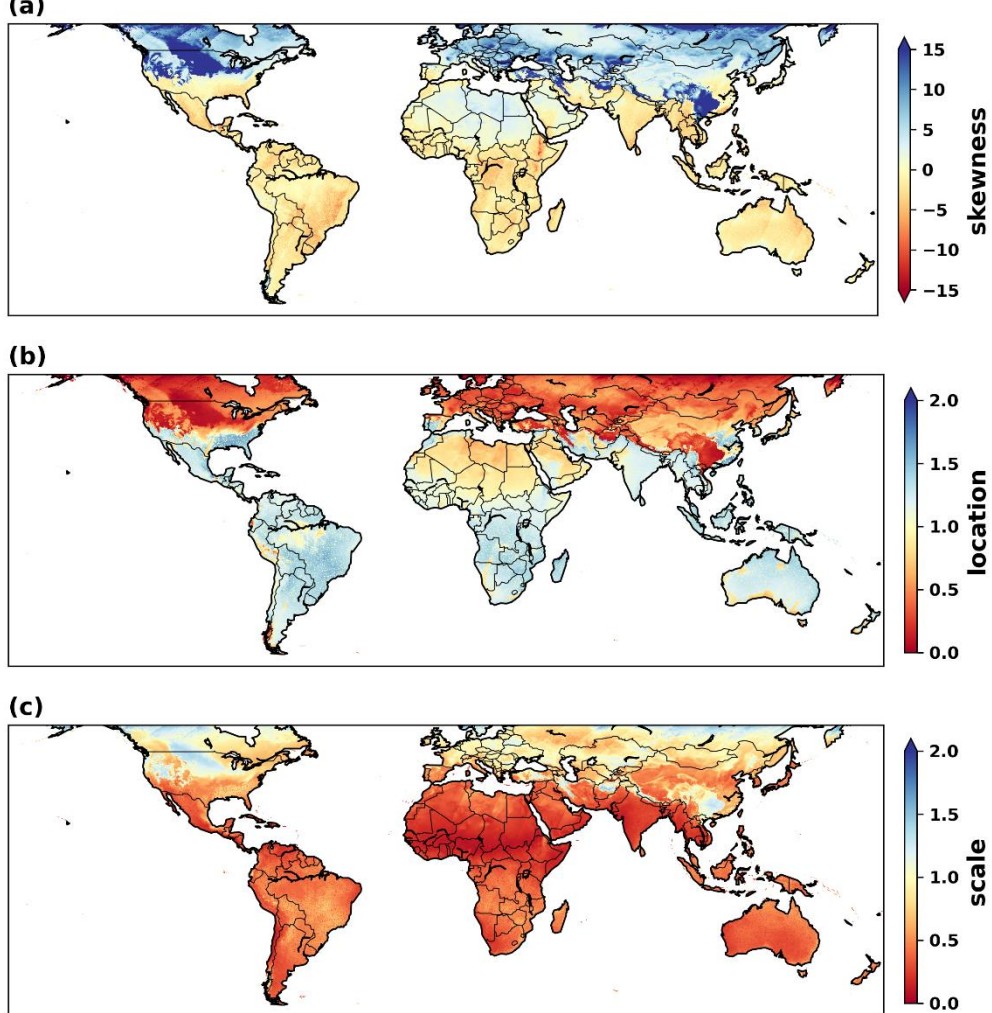

**Figure 3: The parameters representing the noise ratio (a) skewness, (b) location, and (c) scale for the month of January over the spatial domain of the stoPET model.**

By way of a worked example, Fig. 4a shows the monthly distribution of the noise ratio for a single location in Wajir (Kenya), while Fig. 4b shows the randomly generated noise ratio array for January and the parameters representing it. The steps followed to create these noise ratio values were as follows:

1)  Calculate the average hourly PET for each month from the 40 years hPET data. This gives a characteristic diurnal curve from which we can determine the average hourly PET value for each month (the black line in Fig. 1).

2)  Divide each hourly PET for every day in each month (e.g., Jan 1) by its average from step 1. This gives the noise ratio array (Fig. 4a).

3)  Fit a skewed normal distribution to the noise ratio array based on Eq. (2) for each month and save the parameters (Fig. 4b).

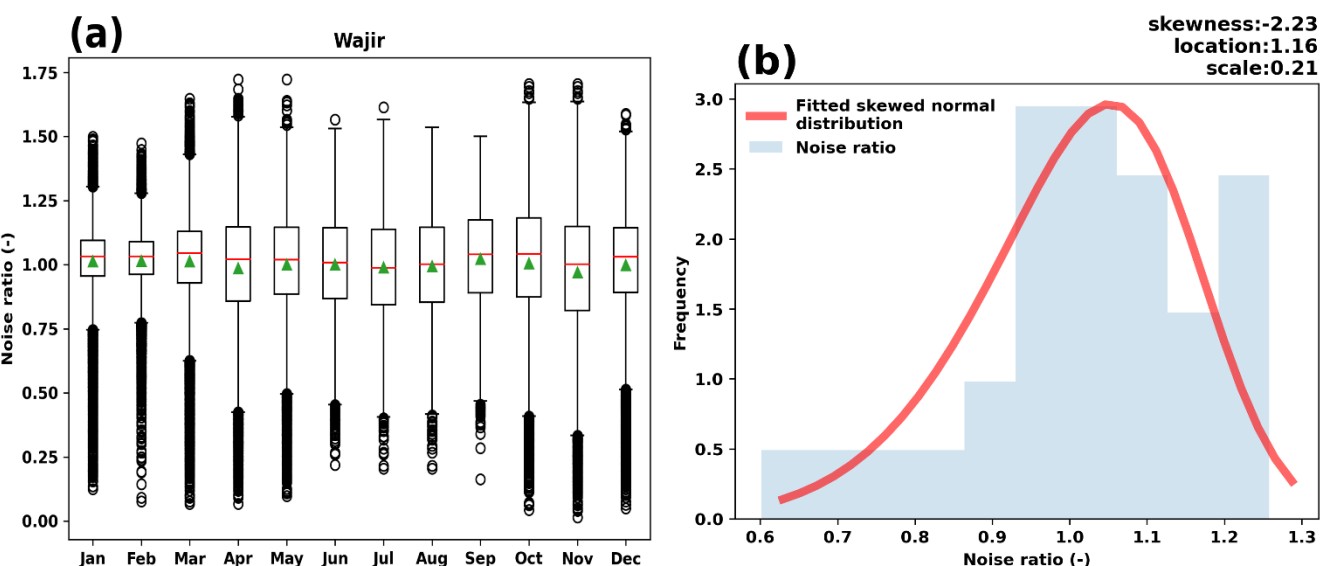

**Figure 4: (a) Noise ratio box plot for a single location in Wajir (Kenya). The box plots indicate that the noise ratio is variable over each month with the green triangle showing the mean and the red line in the box plot indicating the median (b) A histogram for the January noise ratio is shown in blue shaded bars, with the fitted skewed normal distribution shown in red solid line. The corresponding distribution parameters are indicated in the top left of the plot.**

### 2.2.3    Generating stochastic hourly PET

stoPET generates simulated stochastic PET values for a particular month by multiplying the respective sine function (Fig. 1) by the noise ratio sampled from the corresponding skewed normal distribution (Fig. 4b). For instance, for a particular simulation of January PET, stoPET will generate 31 random noise ratios, producing 31 diurnal cycles of PET that amplify (or dampen) the mean diurnal PET sine wave for the month. Synthetic PET for any climate scenario can then be generated for the entire month and for as many years as the user chooses.

## 3    Model verification

### 3.1.1    Verification of stoPET against hPET dataset

*Regional representation*

stoPET is set up to generate synthetic plausible hourly PET time series within any defined spatial area between 55° N and 55° S. High latitude areas were not included because some months do not have a clear sunset and sunrise times during summertime creating potential errors in the sine function fitting. We have evaluated the stoPET model against hPET (the only globally available dataset at hourly resolution) (Singer et al., 2021) and against the Climate Research Unit's daily average PET dataset generated by the PM method at monthly temporal resolution (presented as a daily average for the month) over the period 1901-2018 at 0.5° grid resolution (CRU,   (https://crudata.uea.ac.uk/cru/data/hrg/,   (Harris et al., 2020)). We carried out these evaluations for selected humid and arid regions on six continents (North America, South America, Europe, Africa, Asia, and Australia [Australia sub-continent also includes the Oceania region]). As an illustration of the visual comparison to hPET, Figure 5 shows the average annual PET climatology for Africa over five years of simulated PET from stoPET (Fig. 5a) against five randomly selected years from the hPET dataset, where we have also removed the nighttime PET values (Fig. 5b) since stoPET considers the nighttime PET to be zero. Figure 6 shows a similar comparison for Europe (stoPET, Fig. 6a and hPET, Fig. 6b). These comparisons indicate that stoPET estimates annually averaged PET values from hPET with only an average

percentage difference of ~±5 % (see Fig. S3 to Fig. S8 in the supplementary document). The results of this comparison, albeit qualitative, suggest a strong similarity in globally distributed values between the simulated and historical data in most regions of the world, which supports the use of stoPET for representing the annual PET over large regions. The figures for the remaining continents are provided in the supplementary document (Fig. S3 to Fig. S8), along with the contrast between stoPET annual PET and the hPET dataset when the nighttime values are included (Fig. S9 to Fig. S14).

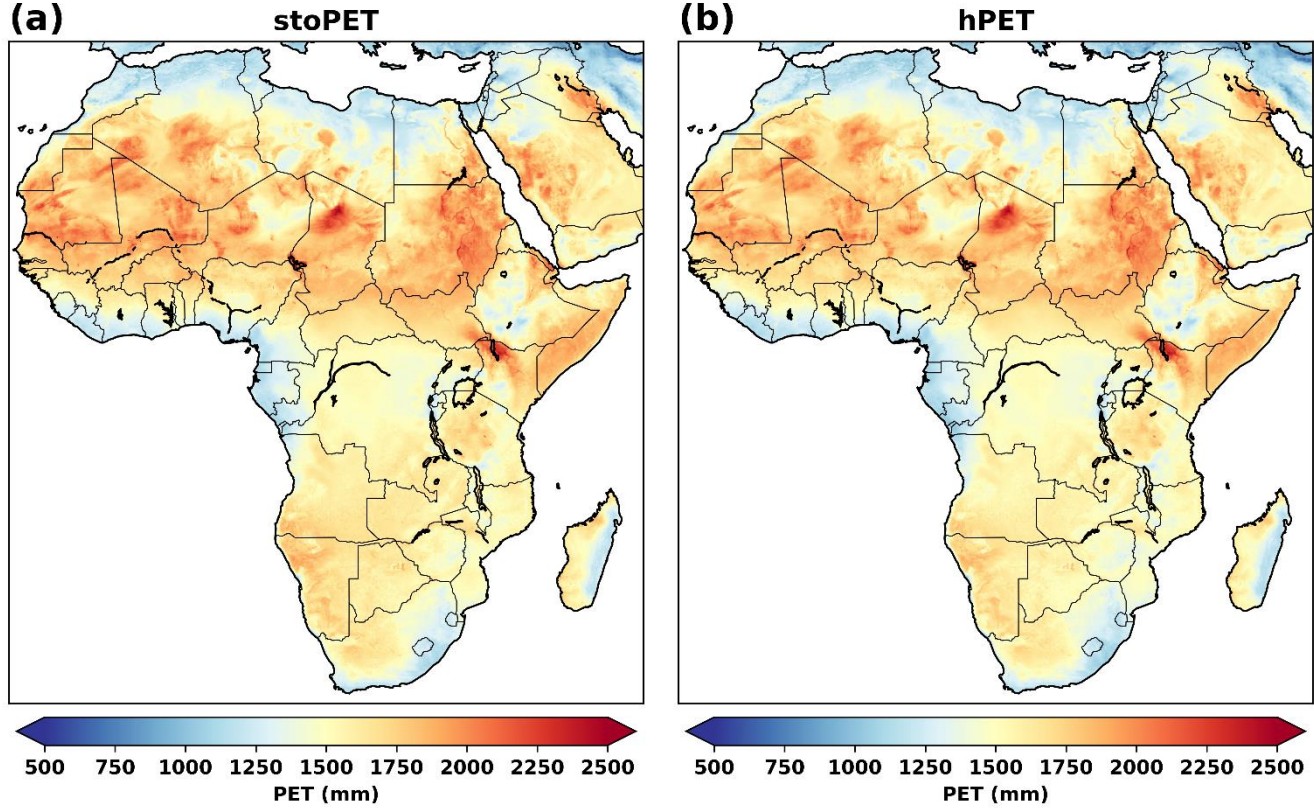

**Figure 5: Average annual PET for five randomly selected years (a) stoPET, (b) hPET with nighttime PET removed for Africa.**

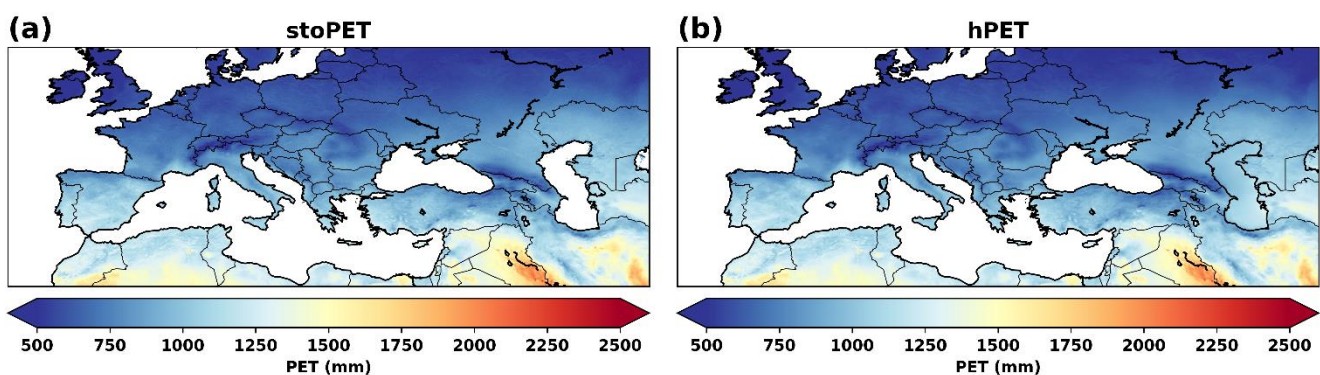

**Figure 6:  Average annual PET for five randomly selected years (a) stoPET, (b) hPET with nighttime PET removed for Europe.**

*Single point representation*

To verify the performance of stoPET more quantitatively, analysis was carried out on twelve points across 6 continents chosen to be representative of both humid and arid climates across the global land surface (Fig. 7). Ten ensembles, each comprising 20 years of synthetic PET data, were generated using stoPET and compared against the hPET dataset over the period 2001-2020, substituting the nighttime (zero) PET values of stoPET with nighttime values of hPET. Next, the hourly PET values from stoPET and hPET were aggregated to daily average PET values for each month at the twelve locations for evaluation of stoPET (again, including the nighttime values), against the CRU PET dataset developed by the PM method (see above).

We carried out three statistical analyses on the monthly aggregated, daily averaged values of stoPET compared to hPET and CRU: (a) pBias (Eq. (3)) indicates whether the stochastically generated values overestimate or underestimate the comparable values of hPET and CRU (b) Normalized Root Mean Square Error (NRMSE) (Eq. (4)), a so-called 'scatter index', measures the similarity of stoPET compared to hPET and CRU datasets. NRMSE is normalized by the mean of each dataset, and (c) a two-sample Kolmogorov-Smirnov test compares the full distributions of two datasets of monthly average PET values against each other (Helsel et al., 2020). The equations for pBias and NRMSE are:

$$pBias = \frac{\sum_{m=1}^{m=n}(Y_m - X_m)}{\sum_{m=1}^{m=n} X_m} \times 100\% \qquad \text{Eq. (3)}$$

$$NRMSE = \frac{\sqrt{\frac{\sum_{m=1}^{m=n}(Y_m - X_m)^2}{n}}}{\frac{1}{n}\sum_{m=1}^{m=n} X_m} \qquad \text{Eq. (4)}$$

Where $X_m$ represents the monthly average PET of hPET or CRU for each month, $Y_m$ is the monthly average PET estimated by stoPET and $n$ is the number of months.

Based on these tests, first we find that PET estimated by stoPET is statistically comparable to hPET historical data (Fig. 8 and Fig. 9). This result was encouraging, if not unexpected, since stoPET was designed to create plausible stochastic realistic simulations of hourly PET using hPET as a template for diurnal and seasonal variations in PET. The pBias values between stoPET and hPET range between 0.49 % to 9.68 %, indicating that stoPET is not systematically overestimating or underestimating PET values relative to hPET (Table 1). The NRMSE values range from 0.02 to 0.1 for humid and 0.02 to 0.04 for arid sites, and NRMSE values are small (<0.1) for all locations, indicating low scatter between hPET and stoPET. The Kolmogorov-Smirnov test also shows that stoPET and hPET have statistically similar distributions (p-values at all locations are greater than the threshold 0.05, Table 1). Finally, stoPET produces PET values that are comparable to hPET in terms of capturing the seasonal cycle and variability (Fig. 8 and Fig. 9).

Previously, CRU PET estimates were found to be comparable to hPET values (Singer et al., 2021). Here we directly compare the stochastically generated PET values from stoPET against estimated independent PET values from CRU to evaluate whether stoPET captures the seasonality and mean behaviour within CRU. The comparison between stoPET and CRU indicates that, except in two humid locations (H2 and H6), stoPET values are statistically similar to the independent CRU PET values (Table 1). Even though the pBias and NRMSE values from comparisons between stoPET and CRU are higher than for the hPET comparisons, the p-values of the Kolmogorov-Smirnov test show that stoPET has a similar statistical distribution as CRU for most of the comparisons (except for two humid sites, H2 and H6, which had lower and higher CRU PET values, respectively, within overall narrow distributions). Additionally, stoPET well captures the seasonality of the CRU PET (Fig. 8 and Fig. 9). These evaluation steps give us confidence that stoPET is generating PET (on a monthly timescale) that is largely consistent with existing data products and can therefore be considered as a useful simulator of PET at the global scale.

# Selected locations for point analysis

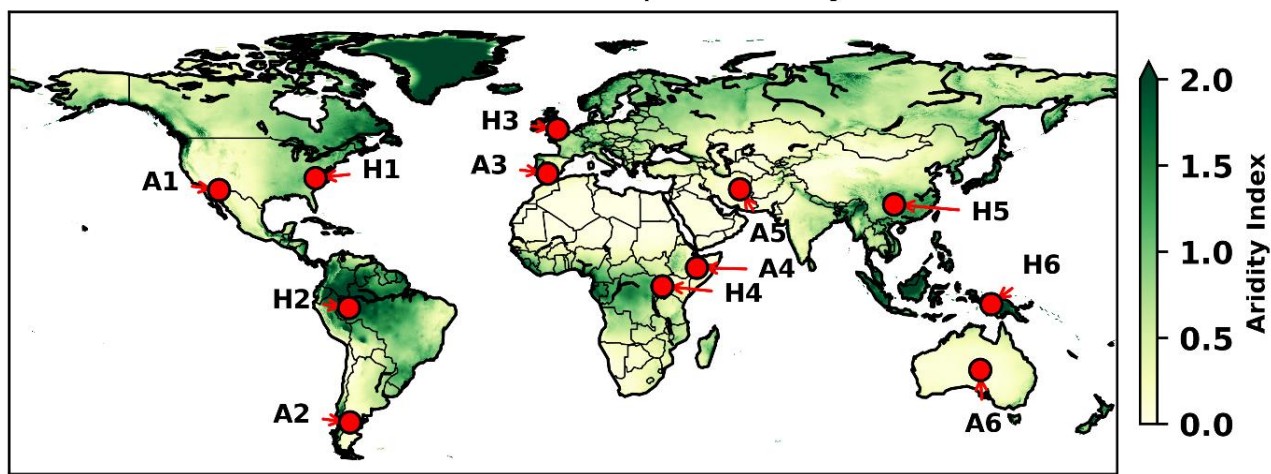

**Figure 7: Single point locations selected for global evaluation for humid and arid climate locations based on the Aridity index data from Consultative Group on International Agricultural Research (CGIAR) (Trabucco and Zomer, 2018).**

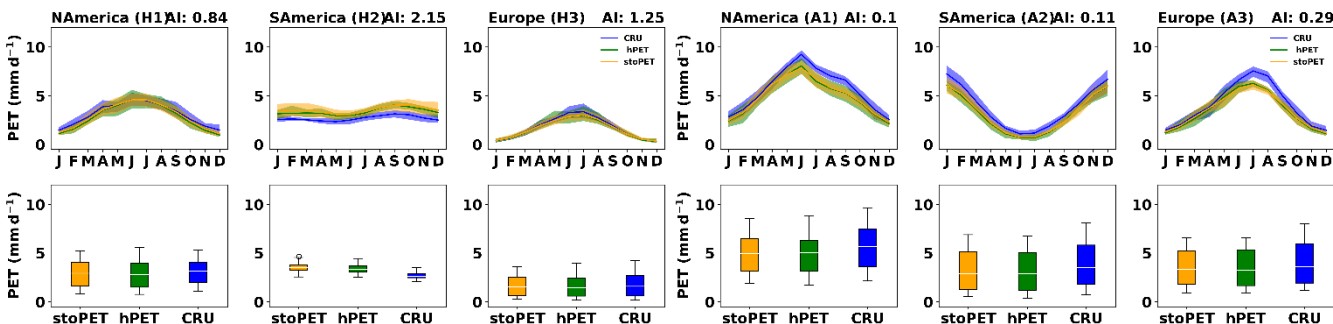

**Figure 8: The seasonal PET and box plots for three datasets over North America, South America, and Europe for the (a) humid and (b) arid locations as located in Fig. 7. The box plots show the distribution of each dataset over the 20 years period. The box indicates the IQR ($25^{th}$-$75^{th}$) while the upper whiskers are set to ($75^{th}$ + 1.5 * IQR) and the lower whiskers are set to ($25^{th}$ - 1.5 * IQR).**

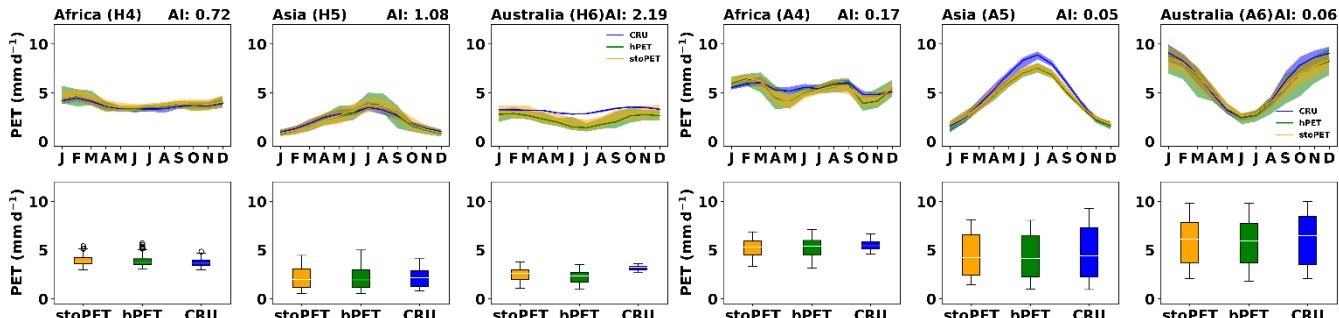

**Figure 9: The seasonal PET and box plots for three datasets over Africa, Asia, and Australia (including Oceania region) for (a) humid and (b) arid locations as located in Fig. 7. The box plots show the distribution of each dataset over the 20 years period. The box indicates the IQR ($25^{th}$-$75^{th}$) while the upper whiskers are set to ($75^{th}$ + 1.5 * IQR) and the lower whiskers are set to ($25^{th}$ - 1.5 * IQR).**

**Table 1: The pBias, NRMSE and Kolmogorov-Smirnov test values between stoPET and hPET as well as between stoPET and CRU for the humid and arid locations on 6 continents as indicated in Fig. 7**

| | pBias (%) | | NRMSE (-) | | KS-stat | | P-value | |
|---|---|---|---|---|---|---|---|---|
| | hPET | CRU | hPET | CRU | hPET | CRU | hPET | CRU |
| **H1** | 3.44 | -5.27 | 0.04 | 0.08 | 0.17 | 0.17 | 1.00 | 1.00 |
| **H2** | 4.31 | 31.64 | 0.05 | 0.32 | 0.50 | 0.92 | 0.10 | 0.00 |
| **H3** | 1.96 | -5.37 | 0.03 | 0.10 | 0.08 | 0.17 | 1.00 | 1.00 |
| **H4** | 1.58 | 5.56 | 0.02 | 0.06 | 0.25 | 0.33 | 0.87 | 0.54 |
| **H5** | 2.48 | -0.22 | 0.05 | 0.11 | 0.17 | 0.17 | 1.00 | 1.00 |
| **H6** | 9.68 | -21.95 | 0.10 | 0.25 | 0.42 | 0.75 | 0.26 | 0.00 |
| **A1** | 0.87 | -12.45 | 0.02 | 0.15 | 0.08 | 0.25 | 1.00 | 0.87 |
| **A2** | 1.86 | -16.29 | 0.03 | 0.17 | 0.17 | 0.25 | 1.00 | 0.87 |
| **A3** | 0.77 | -14.10 | 0.04 | 0.18 | 0.17 | 0.25 | 1.00 | 0.87 |
| **A4** | 0.49 | -3.91 | 0.03 | 0.10 | 0.17 | 0.33 | 1.00 | 0.54 |
| **A5** | 1.89 | -9.40 | 0.02 | 0.14 | 0.17 | 0.25 | 1.00 | 0.87 |
| **A6** | 0.84 | -4.43 | 0.02 | 0.08 | 0.08 | 0.17 | 1.00 | 1.00 |

We carried out additional analyses to evaluate hourly stoPET values against to the native resolution of hPET. Here we only show the results from a single location (point A1 in Fig. 7) as an example; however, the results and plots of the other locations are provided in the supplementary material (Fig. S15 to Fig. S25). The scatter plot (Fig. 10a) indicates that stoPET generates hourly PET values that are comparable to hPET (R = 0.83). The box plots (Fig. 10b) show that stoPET also produces a comparable mean (green triangle in Fig. 10b) and median (red line in Fig. 10b) to hPET and captures the overall variability in the hPET distribution. Figure 11 shows the density plots of the hPET and stoPET data, which indicates that the randomly generated stoPET values well represent the hPET data for the arid location in North America (and other locations, see supplemental figures). Additionally, we investigated how well stoPET captures the diurnal cycle contained within hPET. Figure 12 shows an hourly time series for 15 days of stoPET and hPET over several diurnal cycles, demonstrating good consistency in the timing of peaks and troughs, but with clear evidence of the desired stochasticity in the simulated series.

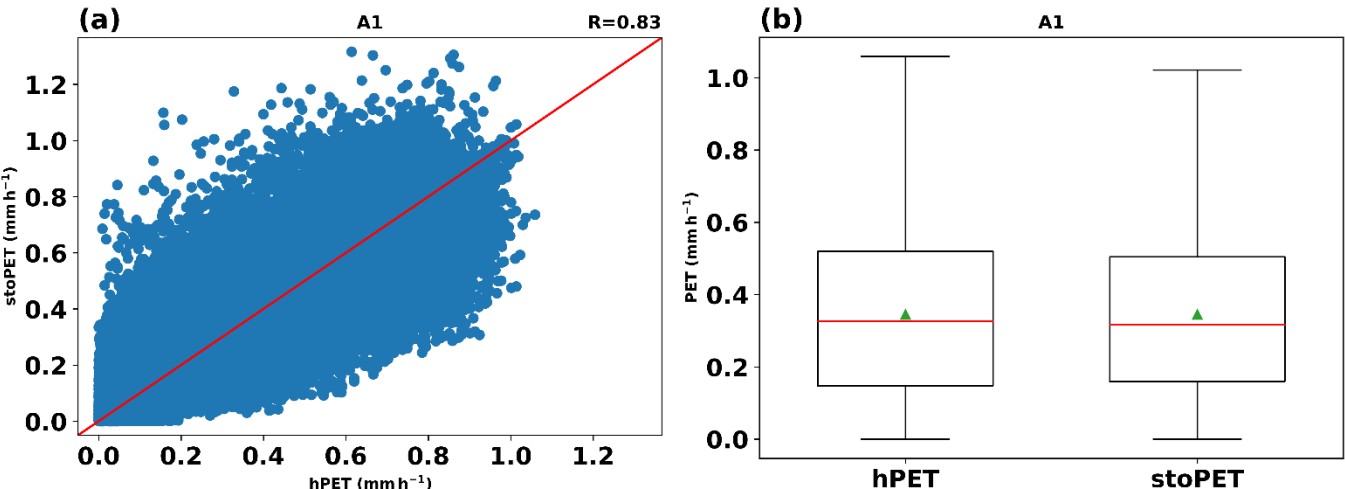

**Figure 10: (a) Scatter plot between hPET and stoPET daytime values, (b) box plots for hPET and stoPET daytime data (green triangle shows the mean and the red solid line indicates the median) over the period 2001–2020 (for A1 in Fig. 7).**

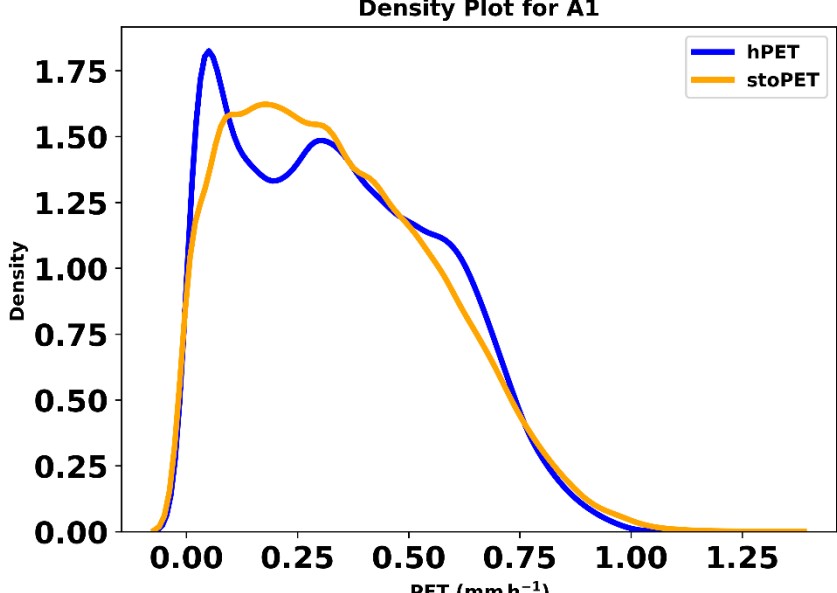

**Figure 11: Density plots for hPET and stoPET for the arid location in North America (for A1 in Fig. 7). The data represent the daytime hourly PET from 2001 to 2020.**

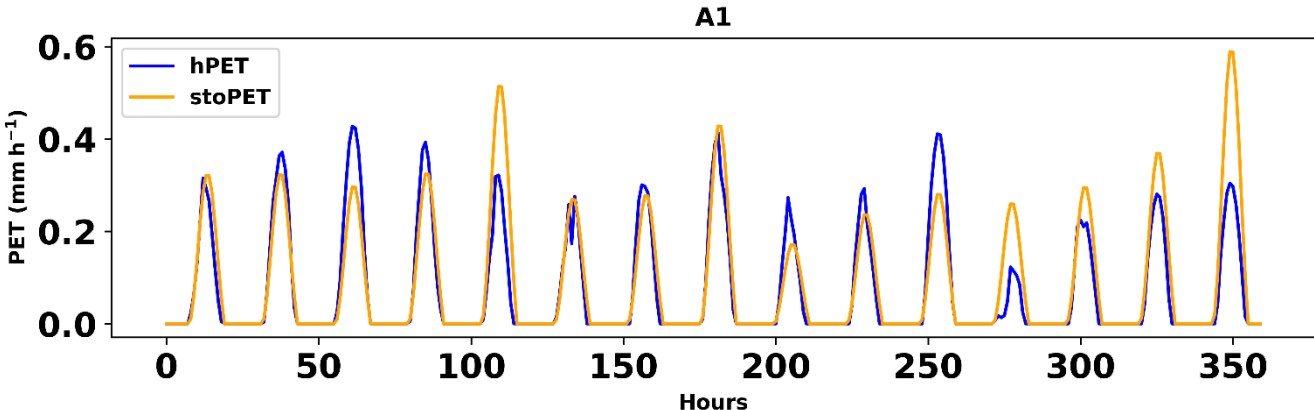

**Figure 12: Time series of hPET and stoPET data for the last 15 days of 2020 (for A1 in Fig. 7). The figure indicates that stoPET capture the diurnal cycle of PET and the difference among each day is an expression of the stochasticity of the model.**

## 4 Incorporating future climate change in stoPET

The future atmosphere is predicted to be warmer due to anthropogenic forcing (Hoegh-Guldberg et al., 2018, IPCC, 2021). This increased atmospheric temperature should lead to higher evaporative demand, which can have substantial impacts on the water balance. stoPET incorporates three methods to account for changes in atmospheric evaporative demand to climate change, supporting flexibility in simulating various climate scenarios. The three methods described below with examples, provide choices for users to explore what fits their study goals.

### 4.1 Method descriptions

#### 4.1.1 Method 1: User-defined percentage step change of annual PET

For some applications, it may be useful to assess the impact of a percentage change in evaporative demand on the water balance. Method 1 consists of the user providing a percentage, corresponding to the desired fractional change in annual PET relative to the historical baseline series (user-defined percentage value - U). This then influences the generation of hourly PET in stoPET as follows:

1) Generate a stoPET series based on historical baseline climate and calculate the annual sum of the simulated series ($PET_{annual}$).

2) Estimate the annual PET change ($\Delta PET_{annual}$) using Eq. (5):

$$\Delta PET_{annual} = PET_{annual} * U \qquad\qquad \text{Eq. (5)}$$

3) Divide $\Delta PET_{annual}$ into monthly changes by multiplying with the average monthly percentage contribution to $PET_{annual}$, which are already generated within stoPET for historical climatology.

4) Divide the monthly change by the number of days in each month to obtain a daily adjustment of the stoPET series.

5) Divide the daily PET change using the percentage contribution of daytime hours, which are calculated within stoPET for each month.

6) The adjusted hourly PET is then obtained based on the summation of the PET from step 1 and the hourly changes of PET from step 5.

### 4.1.2 Method 2: Step change in PET based on a user-defined change in atmospheric temperature

Climate change is often characterised in terms of a specified rise in atmospheric air temperature (Randalls, 2010), which may vary for different locations across the globe but is typically communicated as a global mean temperature change (e.g., 1.5 degrees of warming based on the Paris Climate Treaty) (Kriegler et al., 2018). We fully acknowledge that PET (especially based on the PM method of calculation) is not only driven by temperature changes but by changes in solar radiation, wind speed and humidity (Xu et al., 2014). Nevertheless, to isolate the influence of temperature alone, we created within stoPET a method to calculate temperature-based changes in PET, with all other non-temperature related variables remaining unchanged. This is simply implemented, transparent and aligns directly with global climate discussions and policies (IPCC, 2013; Blunden and Arndt, 2020; NOAA, 2021). Method 2 accounts for a user-defined temperature change and its propagation into hourly PET, which works as follows within stoPET:

1) We recalculated hPET globally with uniform homogenous air temperature increment of 0.5°C (e.g., 0.5°C, 1.0°C, 1.5°C, 2.0°C, 2.5°C) for every hour, with all other non-temperature related variables remaining unchanged.

2) hPET, which was calculated based on the current temperature with no adjustment, was subtracted from newly calculated PET values containing the temperature adjustment. This step revealed that the rate of change of the PET increase is uniform on average (Fig. 13); hence we can use the rate of change in PET and the user-defined temperature change as a multiplicative factor to represent the change in annual PET.

Figure 13 shows an example of annual PET change computed for Wajir, Kenya, where the temperature is raised in increments of 0.5°C from the current temperature. The figure shows a linear relationship between the annual change in PET and change in temperature ($R^2 = 0.998$), as an example, every increase by 0.5°C yields ~55 mm of annual PET change for the specified location. stoPET then provides the global annual PET change based on 1°C of warming derived from 20 years of climatology (Fig. 14). These annual PET changes are used as an input and multiplied by the user-defined temperature factor to determine the amount of annual PET change at each grid cell.

Method 2 adjusts simulated hourly PET generated by stoPET in similar ways to Method 1 (i.e., steps 3-6 are the same as Method 1), but the first two steps are altered as follows.

1) Generate an hourly stoPET time series for one year and take the annual sum.

2) stoPET multiplies the annual change in PET associated with a 1°C temperature increase (Fig. 14) by a user-defined temperature change (ΔT).

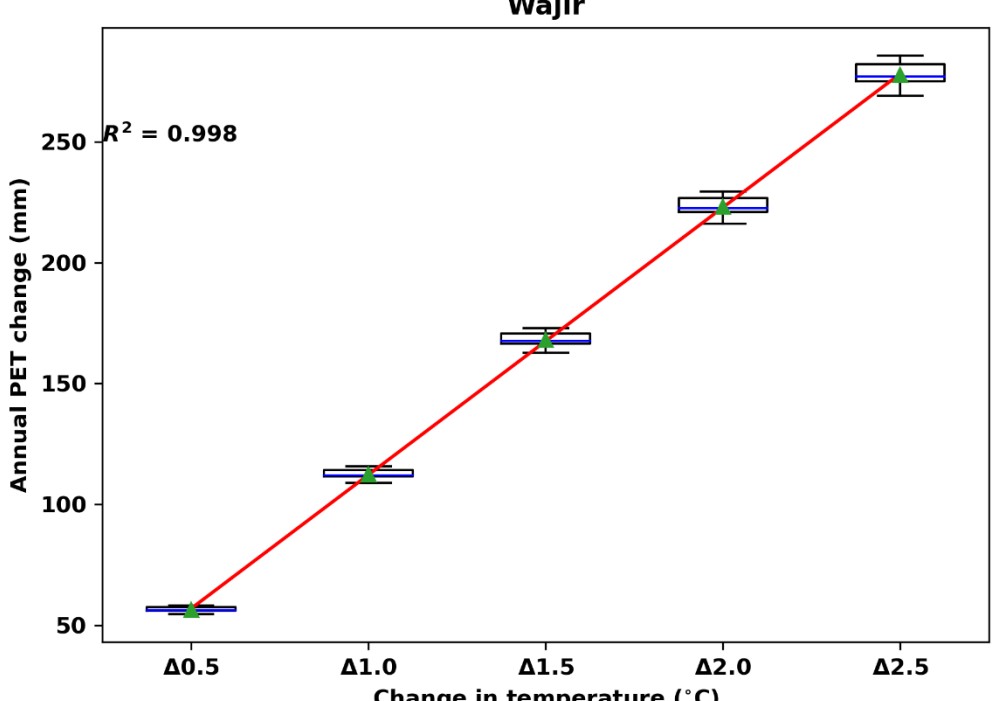

**Figure 13: Annual PET change when estimated by progressively increasing the atmospheric air temperature. The changes are referenced to hPET, which is calculated using the historical temperature. This example is for a single location (Wajir-Kenya). The Red line indicates the regression line with an $R^2$ of 0.998.**

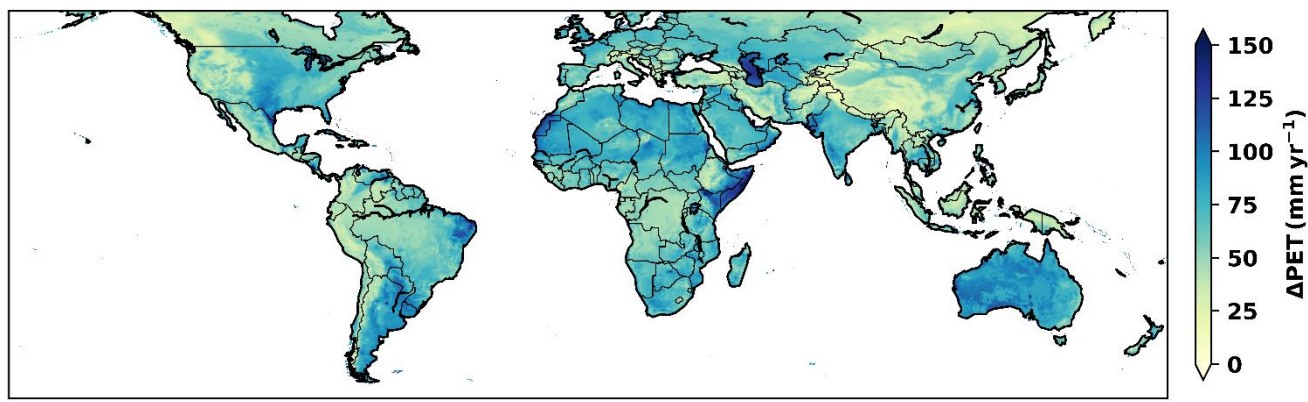

**Figure 14: The global climatological step change in annual PET due to a unit temperature increase. The results were obtained by taking the difference of the PET calculated with increased temperature and hPET which is calculated using the current temperature.**

### 4.1.3    Method 3: Progressive change in PET based on the historical trend in hPET.

In some cases, it may be desirable to evaluate the potential impacts if currently observed trends in PET continue into the future. To support this type of analysis, Method 3 computes the historical trends in hPET for each pixel of the globe and then applies this trend within the stoPET series for every location, leading to progressive change in the simulated PET. stoPET simulates PET via Method 3 as follows, sharing the same steps as Method 1 from step 3 onwards. The first two steps are as follows:

1) Generate stoPET for one year and take the annual sum.
2) Estimate the annual PET change using the slope of the linear trend to historical hPET (Eq. (6)). stoPET computes this trend and uses its slope ($s$) (Fig. S26) as an input parameter applied over the number of years of the simulation ($x$) to adjust the simulated series from stoPET that would be generated based on a 'no climate change' scenario.

$$\Delta PET_{annual} = sx \qquad\qquad \text{Eq. (6)}$$

### 4.2    Examples of stoPET-generated PET under climate change by the three methods

As a demonstration of these methods, we simulated PET under climate changes for arid and humid locations used for model evaluation (Fig. 7). Specifically, we present time series of annual PET for a 5 % user-defined percentage increase in PET (Method 1), a user-defined 1.5°C increase in temperature (Method 2), and by imposing the historical trend from hPET into the future (Method 3) (Fig. 15; Fig. S27 to Fig. S31). These plots demonstrate the built-in flexibility in stoPET for simulating changes to evaporative demand under climate change. For example, they illustrate that under Method 1, there is simply an elevated simulated time series of PET, while the higher values for Method 2 result from propagating a temperature increase through the calculation of PET, and Method 3 shows a clear trend that departs from the historical mean (Fig. 15).

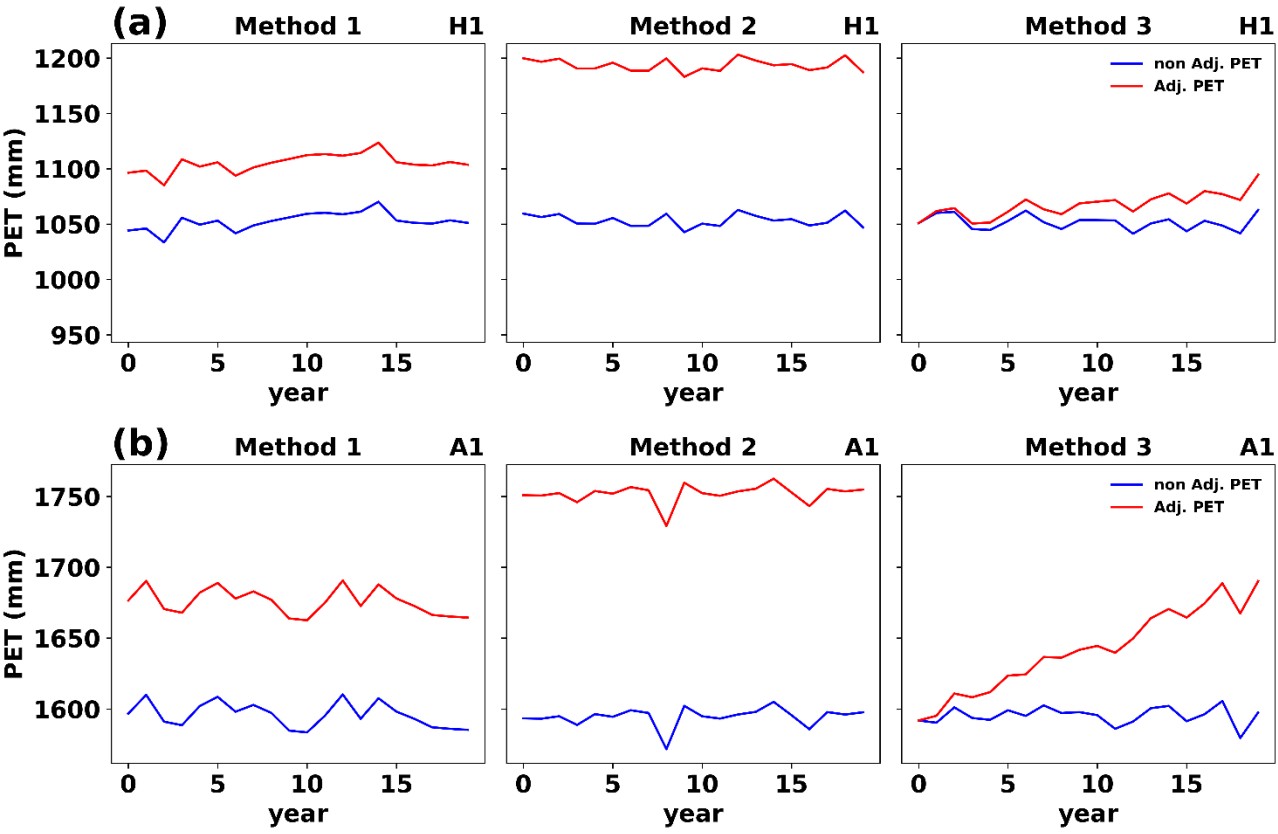

**Figure 15: Annual PET estimated using stoPET with the three climate change methods for (a) a humid location (H1) and (b) an arid location (A1) in North America (see Fig. 7).**

## 5    Discussion

As the global community works to determine the potential impacts of climate change, it is critical to address how changes to atmospheric evaporative demand will affect the water balance and associated water resource availability. Here, we have presented a novel stochastic PET generator (stoPET), which fills a gap in current capabilities to simulate multiple realisations of historical and future evaporative demand across the globe. stoPET is a parsimonious, flexible, and computationally efficient way of generating plausible hourly PET timeseries anywhere on the Earth's land surface for various climatic forcing scenarios. stoPET has the potential for improving climate-related impact studies on the water balance for applications including, but not limited to ecology, ecohydrology, agriculture, and water resources in a wide range of environments across the globe.

The water balance is very sensitive to atmospheric evaporative demand, so the characterization of diurnal and seasonal variability in PET across the globe is a critical component for a wide range of climate impact studies. stoPET is particularly relevant for the prediction of water resource availability, estimation of future crop water demand, assessment of flash flood

risk, and provision of actionable information on expected climatic impacts on the water balance. Given inherent uncertainties in climatic drivers of the water balance (rainfall and PET), simulated assessments of the water balance under potential future climate change would be best framed in a probabilistic way. Stochastic weather generators may provide projections of rainfall and temperature (Chen et al., 2012; King et al., 2015; Steinschneider et al., 2019), but there is currently no standardised capability to simulate plausible time series of PET under a range of future scenarios. It is also not currently possible to retrospectively assess the impact of climate forcing on the historical water balance based on PET. This information gap on PET undermines efforts to drive hydrological, agricultural, and land surface models. We provide a few potential applications of stoPET in this context below.

PET significantly influences the partitioning of the long-term water balance into different stores and fluxes that vary over time and space (Bai et al., 2016; Quichimbo et al., 2021). Key water balance components, including groundwater storage, evapotranspiration, runoff and streamflow are challenging to assess without accurately constraining evaporative demand (Bowman et al., 2016; Condon et al., 2020). An obvious example is flood hazard, which is especially sensitive to antecedent moisture conditions within a drainage basin based on the prevailing PET over the period between rainstorms, which affects the subsequent partitioning of rainfall between infiltration and runoff, the downslope flow of both surface and subsurface water, and correspondingly, the magnitude of flood waves in channels. These influences impact the strength of the watershed response to rainfall events and corresponding flood hazard (Zoccatelli et al., 2019) in a range of environments. stoPET-derived PET will thus support more realistic analyses of the water balance for the purposes of assessing flood hazard (and potential mitigation measures).

Hydrological and land surface models require PET to close the water and energy balance and to resolve its key components (e.g., parsimonious distributed hydrological model for DRYland Partitioning-DRYP, (Quichimbo et al., 2021); PARallel Flow-ParFlow, (Maxwell and Miller, 2005)). Such models are often assessed in terms of the uncertainty in spatiotemporal rainfall used to drive them, but there is additional uncertainty in PET that is typically unconstrained and especially for scenarios of future climate change (Van Osnabrugge et al., 2019). stoPET can generate multiple realisations of PET, supporting the assessment of uncertainty in atmospheric demand and providing key information on PET to support forecasting and risk assessment associated with water availability and agricultural water demand, especially for a wide range of meteorological conditions (Dimitriadis et al., 2021). The stoPET model fills this gap by providing physically realistic PET time series that vary in space and honour the inherent diurnal and seasonal variability.

Water availability to plants is one of the limiting constraints for crop production and food security (Funk et al., 2008; Funk and Brown, 2009; Kang et al., 2009; Ayyad and Khalifa, 2021), but also for the health and functioning of the vegetative ecosystem in natural settings (Mayes et al., 2020; Sabathier et al., 2021; Warter et al., 2021). Forecasts of crop water requirement and irrigation demand for major crops like maize, barley, and wheat (Ewaid et al., 2019) are paramount for preparing advisories related to the timing of planting, crop choice, and irrigation scheduling, especially in arid and semi-arid regions, where high atmospheric evaporative demand and erratic rainfall make farming a risky economic activity (Nyakudya and Stroosnijder, 2011). Crop models require estimates of PET to quantify how much water can be lost to the atmosphere over the diurnal cycle and over the entire season of crop growth. In natural settings, PET is necessary to predict both water availability to plants and the timing of plant phenology, including the timing of green-up and senescence cycles, which have broader implications for ecosystem functioning in a range of environments. In this context, stoPET can be used to simulate the PET and thus assess the hourly availability of water in the soil and its variation over the growing season for a wide range of plants. Our new model also supports analyses of future climatic changes and their impact on natural and agricultural plants, as well as irrigation demand for major crops.

Finally, stoPET can potentially be used in concert with rainstorm generators such as the STOchastic Rainstorm Model (STORM) (Singer et al., 2018), wherein rainfall and interarrival times are simulated to obtain inputs to other models. Rainfall and PET may be straightforwardly interlinked such that PET in stoPET is reduced (due to cloud cover and high humidity) on any simulated rainy day in STORM, thus lowering evapotranspiration losses during rainy periods. In this way, STORM and stoPET would provide consistent sequences of raw data required to close the water balance in terms of key climatically derived variables.

In these and other applications, stoPET presents a new and useful tool to support decision making. For a range of practical situations ranging from water resource planning to agriculture to disaster risk reduction it would be useful to explore the plausible range of variability in PET and its impact on the water balance for any region. For example, the Horn of Africa drylands region is currently experiencing its 5th consecutive season (October-November-December 2020;  March-April-May 2021;  October-November-December 2021;  March-April-May 2021; and  October-November-December 2022) of drought in which atmospheric temperatures are elevated (FEWS NET, 2022a). A 6th 'failed rainy' season is predicted for the upcoming 'long rains' season (March-April-May 2023) (FEWS NET, 2022b). Once a temperature forecast is issued for the region, this information could be used to create multiple stochastic series of PET from stoPET, which could then be used with rainfall forecasts to drive hydrological models. Thus, one could examine what impact these elevated temperatures, alongside forecasted rainfall deficits, would have on water resources, crop yields, and available pasture lands for millions of rural people. The output from such this modelling could then support forecast-based financing decisions, as well as to plan disaster response across this vulnerable region.

Other future improvements of the model that we envisage may be to incorporate other variables apart from temperature change that are likely to be non-stationary and affect PET, such as radiation and wind speed. Additionally, the noise factor sampling used to perturb the stochastic PET is currently independent of adjacent grid points, so there is essentially no spatial autocorrelation, which may be undesirable. The impact of this on the realism of the output is not known a priori. Therefore, applying spatial smoothing to the stoPET output across a grid of simulated values might be a potential future improvement of the model.

In summary, stoPET generates stochastic hourly PET across the globe at high spatial resolution and can estimate future PET under a range of potential future climate changes. The model can be used to evaluate different land surface and water balance models, which are used to predict water availability and other metrics related to the impacts of climate on sectors like agriculture and water use.

## 6 Code availability

The stoPET-v1.0 model python script, the required input files and the user manual are available as open access software and documentation on figshare (10.6084/m9.figshare.19665531).

## 7 Authors Contribution

KM, MBS, RR, MC and DTA developed the idea, DTA wrote and tested the model code, DTA wrote the paper with input from MBS, RR, DM, MC, EQM and KM. MFRG tested the model code.

## 8 Competing interests

The authors declare that they have no conflict of interest.

## 9 Disclaimer

The authors take no responsibility for the use or misuse of the provided code.

## 10 Acknowledgements

We acknowledge financial support from the Royal Society (Grant CHL\R1\180485), the European Union's Horizon2020 Program (DOWN2EARTH, Grant 869550), the United States Department of Defence Strategic Environmental Research and Development Program (Grant RC18-C2-1006), the Natural Environmental Research Council (Grants NE/ R004897/1 and NE/P017819/1), the National Science Foundation (Grants EAR-1700555 and BCS-1660490), the European Research Council (Grant 715254), Natural Environment Research Council (grant NE/R004897/1) and the International Atomic Energy Agency of the United Nations under the Coordinated Research Project (CRPD12014).

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
