# Peer review of "stoPET v1.0: A stochastic potential evapotranspiration generator for simulation of climate change impacts"

_Geoscientific Model Development, 2022_

## Author Comment (AC1)

**Author response to the comments from referee #1**

**General comments response:**

| S/N | Page number | Line number | Referee comment | Correction made / Sentence added to the paper | Response |
|-----|-------------|-------------|-----------------|-----------------------------------------------|----------|
| 1 | 6 | 8 | A) What is the format of skewed normal distribution? Please express the distribution.

 B) Why do you choose distribution? Is that common for PET? If yes, include the references. Otherwise, statistical test must be performed to ensure the distribution.

 C) Is it possible that noise ratio can be negative? If not, other distribution must do better job such as gamma. | | A) We used a Python function to estimate the skewed normal distribution equation. Where the PDF is given by : Skewnorm.pdf(x,a) = 2.*norm_pdf(x)*norm_cdf(a*x) Where, a= skewness and x is the random variable.

 B) We chose the skewed normal distribution because the PET values from which the noise ratio is generated resemble a normal distribution with peak values a little later (skewed) after noon local time (see Fig. 1). Given that the mean values are skewed (see Fig.1 gray shade and red line), we use the skew normal distribution at each grid point |

| | | | | | |
|---|---|---|---|---|---|
| | | | | | and for each month of the year.

C) The noise ratio is always above zero (positive), as it is calculated based only on the daytime PET values from hPET. Ultimately, we found that the skewed normal distribution fit the data the best (Fig 4b). |
| 2 | 3 | 15 | There is no full equation that explains the stochastic simulation model of PET including sine +noise+annual variability. Each element is explained in separate sections. Combined model description must be provided. | The overall stochastic PET generation model can be expressed as follows:

Stochastic PET = (Average diurnal cycle of PET using a sine function * a random Noise ratio)  + user-defined annual PET variability

Each of the three components are described in detail in the subsequent sections. | A sentence has been added to the paper expressing the overall equation/method for simulating PET |
| 3 | 7 | 15 | The overall comparison between hPET and stoPET is not acceptable since the hPET was employed to build the stoPET model. naïve or other stochastic model must be used for comparison. | | The comparison with hPET is necessary in order to show that the patterns and distributions of the stoPET model-derived PET replicate the underlying data that they are supposed to represent. This is a |

| | | | | | crucial point for checking that the model works as it should. . There are no other global stochastic PET models that would be considered 'naïve' in this case and that we could compare against. More importantly, we have already done a detailed comparison between hPET and various other PET products in Singer et al., 2021. |
|---|---|---|---|---|---|
| 4 | 9 | 6-10 | Double cycle of seasonal variability shown in Africa of A4 (Figure9) does not seem perform good. Please describe the potential reasons. | The pBias values range between 0.54 % to 7.76 % indicating that stoPET is not systematically overestimating or underestimating PET values relative to hPET (Table 1). The NRMSE values range from 0.02 to 0.08 for humid and 0.02 to 0.04 for arid sites. The NRMSE shows small values (<0.1) for all locations indicating a good representation of the hPET dataset by stoPET. | Thank you for pointing out this discrepancy. We did some further checks and indeed we found a bug in our code for preparing the data for plotting Figure 8 and Figure 9. The nighttime PET values from hPET were not removed properly, leading to incorrect plots, showing notable discrepancies between hPET and stoPET simulations. We have replaced these two figures using the updated code and provide a sentence summarizing the match between these datasets. |

| | | | | | Figure 8 and Figure 9 are now corrected, and all the statistical analysis (Table 1) are corrected. |
|---|---|---|---|---|---|
| 5 | 17 | 1 | Explanation of the program and data must be provided. Provide specific steps to download the data. | | The step-by-step guide for downloading and generating stochastic PET is provided in the User Guide Manual submitted with our manuscript. See supplemental documents. |
| 6 | 11 | 5 | Fig12: stoPET is the stochastic simulation model. One might have wrong implication that the model was not performed good. Separate panels can be used instead of overlapping. | **Figure 12. Time series of hPET and stoPET data for the last 15 days of 2020 (for A1 in Fig. 7). The figure indicates that stoPET captures the diurnal cycle of PET. The differences between the diurnal curves illustrates the stochasticity of the model, which is a strength of the modeling approach.** | We feel the reviewer may have misunderstood what we are presenting here. We put the timeseries plot on the same panel to indicate that stoPET captures the daily diurnal cycle of PET, which is similar to values from hPET. However, stoPET is a stochastic model, where the noise ratio is chosen randomly, so it may simulate both higher and lower values than the underlying data. This is a strength rather than a weakness of the model.

A sentence has been added to the figure caption to avoid ambiguity about model performance. |

| 7 | | | For example, Method 1 and 2, isn't it better with different user-defined-changes at each year. This reviewer suggest that the authors reasonably set up the scenario to change the annual variation. | | Thank you for this very useful suggestion. We will implement this as a new method of accounting climate change in a subsequent version of the model. However, we do feel that the current options for simulating climate change give the user suitable flexibility for characterizing different scenarios of future climate. |

---

## Author Comment (AC2)

**Author response to the comments from referee #2**

**General comments response:**

| S/N | Page number | Line number | Referee comment | Correction made / Sentence added to the paper | Response |
|---|---|---|---|---|---|
| 1 | 2 | 33-35 | Provide overview of the potential application of the model in the introduction. | | A sentence is already written in the introduction naming some potential applications of the model (page 2, line 33-35). We also provide further details of potential applications in the discussion. |
| 2 | 8 | | Wouldn't be more interesting a comparison with a different stochastic source? E.g. Hargreaves computed PET with input from a stochastic weather generator (at higher computational cost)? | | What we wanted to show in Fig. 5 is that the stochastic PET generates PET values that are consistent with separately calculated PET. Comparing it with other stochastic PET formulations is not possible because we are aware of no other global stochastic PET generator.

Furthermore, other methods for PET estimation (e.g., Hargreaves) are not directly comparable to values generated by the Penman-Monteith method. |
| 3 | 14 | 1 | About method 3, adoption of linear trends for timeseries of complex variables can hardly be considered robust. | | Thank you for pointing out this.
The model provides three options to modify PET, of which the use of linear trend is one. The idea is to provide users more flexibility to generate stochastic PET which accounts for potential future changes. In our analysis of hPET we recognized that many locations |

| | | | | | |
|---|---|---|---|---|---|
| | | | | | exhibited linear trends (associated with increases in atmospheric temperature), so providing the option for a linear trend seemed sensible.

We are considering adding additional methods in subsequent versions as suggested by another referee. One such method would account for year-to-year variable temperature changes rather than using a single value of step change in temperature for all years.

The historical hPET data used is the paper is a 40-year long record (1981 to 2020). However, hPET is updated till 2021 now. |
| | | | 4.1.3 when do you consider the beginning for the historical PET start and how long is it? | | |

---

## Referee Report (RR1)

Asfaw et al. (2022) proposed a useful and simple stochastic PET model by perturbing an existing high-resolution PET dataset. The framework has wide application on decision-making. Overall, the manuscript is well written.

Major comments

*The author might need to discuss the limitation of using hPET for both input data and benchmarking data. I suggest having an independent benchmarking data for validation (observations or reanalysis). For example, it can be added to Figure7/8 with another observed-based dataset.

*How do you calculate PET when temperature is below freezing point?

Minor comments

Page 2 Line 9: Some of the climate models use PET concept while others not. For example, CLM do not use PET concept. So, it's not just an issue of outputting. (Same for page15, line33)

Section2.2.1: Is there any justification of assuming nighttime PET as 0? There is still ET at night. However, I am not sure whether it is important to your applications.

Section2.2.2-3: Does the noise ratio has an option to include the year-to-year variability?

Figure 4: Does the box plot showing numbers of (31*available hours) noise ratios?

Section2: Could author add a brief introduction of hPET dataset and PM method? I think that would be useful for the reader.

Figure8,9, Table1: it would be nice to have more scientific explanations. It seems to have larger ensemble spread in the drier regions. How is that related to the diurnal cycle, or year-to-year variations. pBias are larger in humid regions and why?

Page 12 line 35: "per hour"-> "for every hour" or "for each given time". The original statement is easily to be misunderstood (e.g., slope or rate of changes)

Section 4.1.3: Do you have a figure for historical trend in PET? What is the calculated period?

Figure 15, method3: What are the slopes? It seems the historical trends are not obvious.

Page35 line 30 (final conclusion): I suggest the author to emphasize the applications of this model to decision making rather except for evaluating land surface or hydrological models.

---

## Author Response (AR2)

**Author responses to the comments from Referee #2**

**Major comments:**

1. Discuss the limitation of using hPET for both the input data and benchmarking. Independent benchmarking data for validation added on Fig 7/8.

**Response:**
Thank you for this suggestion. To address the issue of independent benchmarking, we have added an independent PET dataset from CRU to support comparison between the stoPET-generated PET and a totally independent PET dataset (monthly resolution), which was also estimated using Penman-Monteith method. Fig. 8 and Fig. 9 and Table 1 have all been updated to include the CRU dataset for comparison.

**Relevant text has been added to the abstract and to the Model Verification section:**
*Page 7 line 21, the following has been added:*

We have evaluated the stoPET model against hPET (the only globally available dataset at hourly resolution) (Singer et al., 2021) and against the Climate Research Unit's daily average PET dataset generated by the PM method at monthly temporal resolution (presented as a daily average for the month) over the period 1901-2018 at 0.5° grid resolution (CRU, (https://crudata.uea.ac.uk/cru/data/hrg/, (Harris et al., 2020)).

*Page 8 line 10, the following has been added to reflect the inclusion of the CRU dataset:*

To verify the performance of stoPET more quantitatively, analysis was carried out on twelve points across 6 continents chosen to be representative of both humid and arid climates across the global land surface (Fig. 7). Ten ensembles, each comprising 20 years of synthetic PET data, were generated using stoPET and compared against the hPET dataset over the period 2001-2020, substituting the nighttime (zero) PET values of stoPET with nighttime values of hPET. Next, the hourly PET values from stoPET and hPET were aggregated to daily average PET values for each month at the twelve locations for evaluation of stoPET (again, including the nighttime values), against the CRU PET dataset developed by the PM method (see above).

*Page 9 line 23, the following has been added:*

Previously, CRU PET estimates were found to be comparable to hPET values (Singer et al., 2021). Here we directly compare the stochastically generated PET values from stoPET against estimated independent PET values from CRU to evaluate whether stoPET captures the seasonality and mean behaviour within CRU. The comparison between stoPET and CRU indicates that, except in two humid locations (H2 and H6), stoPET values are statistically similar to the independent CRU PET values (Table 1). Even though the pBias and NRMSE values from comparisons between stoPET and CRU are higher than for the hPET comparisons, the p-values of the Kolmogorov-Smirnov test show that stoPET has a similar statistical distribution as CRU for most of the comparisons (except for two humid sites, H2 and H6, which had lower and higher CRU PET values, respectively, within overall narrow distributions). Additionally, stoPET well captures the seasonality of the CRU PET (Fig. 8 and Fig. 9). These evaluation steps give us confidence that stoPET is generating PET (on a monthly timescale) that is largely consistent with existing data products and can therefore be considered as a useful simulator of PET at the global scale.

2. How did you calculate PET when temperature is below freezing point?

**Response:**

The PET is calculated using the FAO Penman-Monteith equation. The temperature data is from ERA-5 Land and no changes have been made to the equation or the temperature data. The full description of the PET calculation is described in the paper (Singer et al., 2021). The PET is adjusted based on the available air temperature and dew point temperature since the PET is also controlled by other variables like wind and radiation, we will still have a potential evapotranspiration value for the location even though the temperature is below freezing point. However, the values of the PET will be highly reduced.

**Minor comments response:**

1. Some of the climate models use PET concept while others not. For example, CLM do not use PET concept. So, it's not just an issue of outputting. (Same for page15, line33)

**Response:**

We agree and re-wrote the introduction to make it more clear based on this comment from the reviewer.

***Page 2 line 8, the following has been added:***

While some global climate models do not include PET explicitly (e.g., COSMO-CLM (Will et al., 2017)), most global climate models (e.g., ERA5-Land) do provide some of the outputs of climatic variables used to estimate PET. However, they do not directly output PET itself, which would support more detailed impact-based modelling of climate change. Climate models focus on predicting the effects of greenhouse gas emissions on global water and energy transfer, and thus they output climate variables (e.g., temperature, radiation, surface pressure, wind speed, and rainfall). Without explicit data on PET, high computational resources are required to estimate the PET for large areas from climate model output variables, and the spatial and temporal scales of these outputs are typically too coarse for detailed impact analyses. These scaling considerations may make climate model output unsuitable for computing PET. This is especially true for application to certain water balance applications, where diurnal changes in PET are important for a specific location or where there are large spatial differences in PET. Downscaling techniques are commonly used to generate the parameters needed to estimate PET from global climate models by the PM method (or other methods) at the appropriate resolution, but this increases the computational resource requirement (Tukimat et al., 2012) and adds additional uncertainty to PET calculations.

Another challenge for PET estimation is how to characterise evaporative demand under climate change scenarios, which is an important need for assessing possible future climate change impacts (Xu et al., 2014). Temperature is one of the major climate variables influencing PET (Allen et al., 1998). Therefore, with increasing temperature under climate change for most of the globe, there is a need to simulate historical and future PET in a consistent and spatially explicit way. Simulating changes in evaporative demand associated with changes in temperature would be particularly useful for assessing the potential impacts of meeting/not meeting the 1.5° C target of the Paris Climate Treaty (Kriegler et al., 2018) or for addressing any future global temperature target. Additionally, it would be powerful to be able to simulate step changes and trends in PET according to user-defined specifications, giving the user a flexible tool for generating a range of PET time series for various applications.

2. Is there any justification of assuming nighttime PET as 0? There is still ET at night. However, I am not sure whether it is important to your applications.

**Response:**

Nighttime PET is very small and sometimes it is negative, indicating condensation. Nighttime variations in PET have little impact in most applications (e.g., crop or hydrological modelling), as most of the variation in PET

(and the high magnitude of PET) occurs in the daytime due to the prominent role of the Sun. Additionally, there is no clear way to characterise nighttime variations in PET as is possible for daytime values (given the strong diurnal cyclicity). Hence, we set nighttime values of stoPET to zero. However, we note that we have provided new explanation of this in the manuscript text, which also quantifies the nighttime values and presents them in comparison to daytime values within supplemental figures Fig S1 and Fig S2.

***Page 4 Line 3, the following has been added:***

In reality, PET is not always zero at night, but it typically ranges from small positive to small negative values (representing condensation) within hPET. For example, nighttime PET is relatively higher in arid regions (median PET value is between 0.001 and 0.076 mm h$^{-1}$) compared to humid regions (median PET value is between -0.014 and 0.002 mm h$^{-1}$) (Fig. S1 and Fig. S2). Nevertheless, the impact of nighttime PET in core applications such as crop and hydrological modelling is expected to be minimal, hence we set nighttime PET values to zero in stoPET.

3. Does the noise ratio have an option to include the year-to-year variability?

**Response:**

The noise ratio is generated randomly for each month and each year of a simulation. Since it is randomly generated, the values for each year and each month will be different. This point is made in Section 2.2.3.

4. Figure 4: Does the box plot showing numbers of (31*available hours) noise ratios?

**Response:**

Yes, the box plots show the noise ratio for each month and hours for the data period considered (1981-2020), but only for daytime hours, since we nighttime PET to be zero.

5. Section2: Could author add a brief introduction of hPET dataset and PM method? I think that would be useful for the reader.

**Response:**

A full description of the hPET dataset with the detailed PM method used to calculate it is described and published in the Scientific Data journal (Singer et al., 2021). We already specified the method (Penman-Monteith) in the text here and referred the reader to the hPET paper.

6. Figure8,9, Table1: it would be nice to have more scientific explanations. It seems to have larger ensemble spread in the drier regions. How is that related to the diurnal cycle, or year-to-year variations. pBias are larger in humid regions and why?

**Response:**

Thank you for this detailed observation. The remit of this journal is present new models, to evaluate them, and demonstrate their use. It is not to analyze the data and discuss the scientific observations. This is why we did not provide any detailed discussion of these figures. However, we note that arid locations have more seasonality in PET than humid locations, leading to more spread in PET for arid locations. The goal of these figures (and the table) is to show that the stochastic PET generator can produce comparable PET values to that of the calculated PET values from other comparable datasets.

7. Page 12 line 35: "per hour"-> "for every hour" or "for each given time". The original statement is easily to be misunderstood (e.g., slope or rate of changes)

***Page 13 Line 23, the following has been changed:***

'Per hour' was changed to 'for every hour'.

    8.   Section 4.1.3: Do you have a figure for historical trend in PET? What is the calculated period?

**Response:**

A plot of the historical trend of hPET data is now included in the supplementary document as Fig. S26 for the period, 1981 – 2020.

***Page 14 Line 15 the following has been added:***

(Fig. S26) as a reference.

    9.   Figure 15, method3: What are the slopes? It seems the historical trends are not obvious.

**Response:**

The new Figure (Fig. S26) indicates the slope of the historical trend (within the supplementary document).

    10. Page16 line 30 (final conclusion): I suggest the author to emphasize the applications of this model to decision making rather than for evaluating land surface or hydrological models.

**Response:**

We have added a paragraph in the discussion indicating how stoPET can help in decision making.

***Page 17 Line 8 the following is added:***

In these and other applications, stoPET presents a new and useful tool to support decision making. For a range of practical situations ranging from water resource planning to agriculture to disaster risk reduction it would be useful to explore the plausible range of variability in PET and its impact on the water balance for any region. For example, the Horn of Africa drylands region is currently experiencing its $5^{th}$ consecutive season of drought in which atmospheric temperatures are elevated. A $6^{th}$ 'failed rainy' season is predicted for the upcoming 'long rains' season (March-April-May 2023). Once a temperature forecast is issued for the region, this information could be used to create multiple stochastic series of PET from stoPET, which could then be used with rainfall forecasts to drive hydrological models. Thus, one could examine what impact these elevated temperatures, alongside forecasted rainfall deficits, would have on water resources, crop yields, and available pasture lands for millions of rural people. The output from such this modelling could then support forecast-based financing decisions, as well as to plan disaster response across this vulnerable region.

---

## Author Response (AR3)

**Author responses to the comments from Referee #2**

**Minor comments:**

1. User guide: I suggest providing an *.yml file for sharing python environments.

**Response:**
An environment.yml file is now provided at the model script link [10.6084/m9.figshare.19665531](10.6084/m9.figshare.19665531) so users can download and use it. We also provide the relevant information in the user guide manual of the model.

2. Page 7, Line 23: Is " presented as a daily average for the month" all-day average, which should be fair for CRU comparison (without removing nighttime as fig. 5,6)?

**Response:**
Thank you for the suggestion.
Indeed, the CRU is a daily average value of PET for each month. In order to directly compare, we would need to remove the nighttime values from CRU, which we cannot do since they are provided as a daily average value. We wanted to show on Fig. 5 and Fig. 6 that the stochastically generated PET (stoPET) captures the spatial variability of PET from hPET. However, we have included in the point analysis (Fig. 8 and Fig. 9) that by adding the nighttime PET of hPET to stoPET and comparing it with CRU, the stoPET data are within the range of values of widely used (e.g., hPET and CRU).

3. Page 2 Line 8-29: I am grateful the authors clarifying LSM-PET. Is it possible to condense the words? I am afraid current statements are too long.

**Response:**
Thank you for the suggestion.
We modified this paragraph based on some of the comments from the previous review. The paragraph explains the challenges related to PET data availability within LSMs and the importance for the water balance. Although the paragraph seems long, it clarifies the rationale for our work in detail. Condensing it would result in leaving out some sentences and we fear that it might not clearly state what we want to convey. We prefer to leave the paragraph as it is as it makes it more clear for readers.

4. Page 17 line 11, " is currently experiencing its 5$^{th}$ consecutive season of drought" Please add a specific time range here and reference , if possible.

**Response:**
Thank you for the suggestion.
Specific time range and references have now been added.

**Relevant text has been added to the main paper:**
*Page 17 line 11, the following has been added:*

(October-November-December 2020; March-April-May 2021; October-November-December 2021; March-April-May 2021; and October-November-December 2022) (FEWSNET, 2022a).

*Page 17 line 14, the following has been added:*

(FEWS NET, 2022b)